# Psychosocial Work Stress and Occupational Stressors in Emergency Medical Services

**DOI:** 10.3390/healthcare11070976

**Published:** 2023-03-29

**Authors:** Rupkatha Bardhan, Traci Byrd

**Affiliations:** Department of Occupational Safety and Health, Murray State University, Murray, KY 42071, USA; tbyrd@murraystate.edu

**Keywords:** psychosocial job stress, emergency medical services, effort reward imbalance, overcommitment, sleep apnea, rotating shift

## Abstract

Emergency medical service (EMS) professionals often experience work stress, which escalated during COVID-19. High job demand in the EMS profession may lead to progressive decline in physical and mental health. We investigated the prevalence of psychosocial job stress in the three levels of EMS: basic, advanced, and paramedic, before and during the COVID-19 pandemic. EMS professionals (n = 36) were recruited from EMS agencies following the Institutional Review Board approval. Participants took surveys on demographics, personal characteristics, chronic diseases, and work schedules. Job stress indicators, namely the effort–reward ratio (ERR) and overcommitment (OC), were evaluated from survey questionnaires using the effort–reward imbalance model. Associations of job stress indicators with age, sex, body mass index, and working conditions were measured by logistic regression. Psychosocial work stress was prevalent with effort reward ratio > 1 in 83% of participants and overcommitment scores > 13 in 89% of participants. Age, body mass index, and work hours showed strong associations with ERR and OC scores. The investigation findings suggested that a psychosocial work environment is prevalent among EMS, as revealed by high ERR, OC, and their correlation with sleep apnea in rotating shift employees. Appropriate interventions may be helpful in reducing psychosocial work stress in EMS professionals.

## 1. Introduction

Emergency medical service (EMS) professionals are subjected to considerable physical and emotional demands when serving patients suffering from severe health conditions or accident victims in life-or-death situations. Occupational stress within the EMS profession could be attributed to a number of factors such as hostile or hazardous environments, repeated exposure to traumatic situations, physical demands of the occupation, trends of shiftwork, and organization and leadership stressors [1,2]. All of these factors could affect the mental health of the EMS professional and could negatively affect the quality of care provided to their patients [3]. There are gaps in understanding psychosocial stress at work and contributing factors that are associated with stress at work in emergency responders [4]. In this cross-sectional study, we planned to investigate psychosocial stress and risk factors in EMS personnel working in the United States before and during the COVID-19 pandemic.

EMS professionals worked in extremely stressful environments during the response to the COVID-19 pandemic faced by the world. A cross-sectional study from Spain reported the COVID-19 pandemic had a tremendous impact on the mental health of ambulance Emergency medical technicians or EMTs [5]. The study suggested that follow up studies were needed to understand the psychological status of EMTs before and during the pandemic. The study also mentioned deeper investigations on the work conditions of EMTs would be helpful to implement interventions [5]. Although our study on EMS was not planned to be conducted during the time of COVID-19, the study captured the psychosocial stress and risk factors in EMS professionals before the pandemic and during the onset and continuation of the COVID-19 pandemic situation.

A study published among EMS professionals in Texas, U.S. reported changes in prehospital practice during COVID-19, resource shortages that influenced patient care, and a high rate of occupational burnout [6]. Another study on frontline healthcare workers reported stress and burnout during COVID-19 and suggested psychological resources such as resilience and adaptive defense mechanisms are essential in protecting individuals from severe stress and burnout [7]. This study aimed to determine whether psychosocial job stress is prevalent among EMS professionals working in rural and urban environments and whether job stress is associated with other risk factors in the population. Study participants were recruited from EMS agencies in urban and rural Kentucky following IRB approval. Psychosocial job stress was determined from the responses to the online survey questionnaire using effort–reward imbalance model created by Siegrist et al [8].

Emergency Medical Technicians (EMTs) and paramedics have played a significant role as frontline workers during COVID-19 [9]. Their direct contact with patients made them vulnerable to contract coronavirus without protection, thus leading them to psychosocial stress and mental health issues during the unusual times. Further, the EMS professionals already faced many challenges before COVID-19, such as work overloads, rotating shifts, night shifts, and long work hours, which escalated during COVID-19, leading to fatigue and burnout among them [10]. A study published in 2020 showed that British Columbia’s paramedics sought mental health support in April of 2020 at almost twice the rate of the previous year [11]. A spike in calls for help reflected the heightened stress of a job that required emergency responders to assume their patients had COVID-19 [11]. Lack of proper PPE and lack of safety protocols to handle the new virus created fear and stress among healthcare professionals [12]. Burnout is often caused by the chronic workplace stress that is not managed successfully. The combination of persistent stress, intense work demands, inability to manage the situation, and lack of needed resources during COVID-19 created burnout in healthcare professionals [7,13]. Previous studies have shown stress, anxiety, depression, and burnout among EMS professionals during COVID-19 [14,15]. Although stress, depression, anxiety, and burnout have been investigated, there is still a lack of data regarding psychosocial stress and health risk factors that contribute to negative health impact in this group [14]. In addition, a gap exists in understanding the negative impact of working in nightshifts, rotating shifts, extended work hours, and their association with sleep disorders and other health impacts amongst EMTs. This study addressed the gaps on finding psychosocial stress level before and during COVID-19 among EMS personnel and investigated if an association exists between stress, shiftwork, long work hours, and chronic diseases.

Our study helps to understand potential health risks from working in extreme, demanding work environments and ways to adopt measures to reduce job stress. The study is unique in identifying psychosocial work stress in EMS professionals in urban and rural environment before and during pandemic and explores ways to identify stressors associated with work. The discussion that follows focuses on how psychosocial stress and risk factors were assessed and the results, which will provide insight on job stress and risk factors in EMS professionals pre-pandemic as well as in the midst of the pandemic.

## 2. Materials and Methods

### 2.1. Study Design

A cross-sectional pilot study was conducted to investigate the impact of psychosocial stress on EMS in rural and urban Kentucky, U.S. Between October 2019 and July 2020, basic EMTs, advanced EMTs, and paramedics working in EMS agencies participated in our study. A total of 36 participants (n = 36), including both men and women, participated in our study.

We reached out to supervisors of the local EMS agencies to recruit participants for our pilot study (convenience sampling). Out of 64 employees who worked in the local EMS agencies in the area and who were approached for our study, only 36 study participants (56%) responded. Data were collected between October 2019 and July 2020. Perhaps the low number of participant responses may be attributed to the sudden onset of the COVID-19 in early 2020 and the unexpected situation during the pandemic. Unique challenges were faced by the EMS population as frontline workers during the pandemic. Our sample size (n = 36) for the pilot study was in accordance with pilot study sample size guidelines [16]. A study on formula to calculate pilot study sample size was published in the year 2015. The study reported, for a particular problem that might have a given probability of 0.10 or higher occurring in a potential study participant, a sample size of minimum 29 participants may ensure a high level of confidence (i.e., at least 90–95%) for the chosen minimum problem probability [16].

The central limit theorem states that the sampling distribution of the mean will always follow a normal distribution when the sample size is sufficiently large (n ≥ 30) [17]. Previous studies have suggested a minimum of 30 as a sample size is required for the sampling distribution to follow a normal distribution [17]. The Shapiro–Wilk test was conducted using SPSS 29 for normality of the data. Data were found to be normally distributed in our study (ERR *p* value 0.551, OC *p* value 0.421, data tested against normality).

Selection criteria for participants included professionals working in emergency medical services above 19 years of age. Both men and women of different racial and ethnic background were included in the selection criteria. The exclusion criteria for the study were participants below age 19 years, physically or mentally challenged, and pregnant women. There were no such participants in our study who met the exclusion criteria. Recruitment of EMS personnel was conducted by directly approaching the supervisors to send online surveys to their EMS group. The study was conducted in accordance with the Declaration of Helsinki, and the Ethics Committee of University (IRB #19-109, date of approval 4 March 2019) approved the protocol. In this cross-sectional pilot study, EMS professionals (n = 36) gave their informed consent online by clicking survey link following university protocol and agreeing to fill out survey questionnaires. Survey information was collected online anonymously. Study participants were not given the opportunity to provide identification monikers or personal information. The short version of the effort–reward imbalance questionnaire created by Siegrist et al. was given to participants for determining psychosocial work stress. Participants provided information about their work schedule including work hours per week, and whether they work nightshifts and weekends. Age, body mass index (BMI), and sex were also provided by participants in the online survey and were included in the analyses as covariates.

Job stress was evaluated from survey questionnaires using the effort–reward imbalance model identifying two job stress indicators: effort–reward ratio (ER ratio) and overcommitment (OC). Risk factors were evaluated from participant’s demographic characteristics and work schedule questionnaire (Figure 1).

This study design was adopted following Institutional Review Board (IRB) approval to investigate the impact of psychosocial stress on emergency medical services in the United States.

### 2.2. Evaluation of Job Stress Using the Effort–Reward Imbalance Model

Job stress was calculated from the responses given by the study participants in the online survey questionnaires created with survey monkey based on the effort–reward ratio utilizing the effort–reward imbalance (ERI) model, developed by Johannes Siegrist [18]. The short version of the ERI model was utilized, which is comprised of two main scales, extrinsic effort and reward with a four-point response option [19]. Two job stress indicators were identified: effort–reward ratio (ER ratio) and overcommitment (OC). Effort measures the amount of effort a person puts in his/her work and was measured using a 4-point Likert scale (strongly disagree, disagree, strongly agree, and agree). Effort questionnaires were coded as three questions ERI 1–3 with 4-point Likert scale; giving a total score between 3 and 12. Questions on effort were focused on the following topics: constant time pressure due to heavy workload, interruptions and disturbances while performing work, and job becoming more demanding over the past years. A 4-point Likert scale measured reward with 7 questions coded as ERI 4 to 10, a sum score of which varies between 7 and 28. A high ERI score means more effort was given at work and the person received fewer occupational rewards. Questions on reward were focused on receiving respect from superiors, job promotion prospects, job security, and adequate salary. The OC questions captured the essence of personal pattern of coping with work. OC was measured by 6 items coded as overcommitment imbalance or OC 1–6 with 4-point Likert scale and total score can vary between 6 and 24. Questions on OC were focused on topics like overwhelmed by time pressures, relaxed and switched off from work at home, sacrificing too much for the job, and trouble sleeping at night due to concerns over incomplete or undone work. The ER ratio (ERR) is the balance between the effort and reward scales. The ERR captures the imbalance between effort and reward at the individual level. To compute the ER ratio, Effort (E) was in the numerator and Reward (R) was in the denominator, and K was the correction factor that was used to adjust for the unequal number of the effort and reward scores. The ER ratio was calculated by the formula, ERR = K × (E/R), where K = 7/3 correction factor proposed by the ERI model was used in this study [20]. For ER = 1, the person reports one effort for one reward; for ER < 1, the person reports less efforts for each reward; for ER > 1, the person reports more efforts for each reward The effort–reward ratio (range 0.25–4.00) was divided into three categories: low (score 0.25–0.74), intermediate (score 0.75–1.00), and high (score 1.01–4.00). over the overcommitment score (range 6–24) was also divided into three categories: low (score 6–11), intermediate (score 12–13), and high (score 14–24), as suggested by Siegrist et. al. in the effort–reward imbalance model [18,19,20].

### 2.3. Statistical Analysis

Statistical analyses were performed, and graphs and charts were created using SPSS 25 and Graph Pad Prism software. Two types of statistical analyses, descriptive and analytic, were performed. Descriptive statistics included percentage (%), mean, standard deviation (SD), median, and range; analytic statistics included the independent T-test (which is a parametric test of significance) to compare means. Correlation analysis (Pearson correlation) was performed to measure relationship and regression analysis for associations between two variables. Statistical significance was accepted at the 5% level.

## 3. Results

### 3.1. Participant Characteristics

Study participants provided details such as age, BMI, gender, race, chronic illness, years of work experience, and job position. The average age of the participants was 39 years. In total, 72% of participants were male and 28% were female. Descriptive statistics of participants are provided in Table 1.

The survey showed 97% of the participants were white and 3% were Hispanic/Latino. In the study, 36% of participants worked as a basic EMT, 8% as an advanced EMT, and 56% as a paramedic EMT. Most participants had worked as an EMT for 10 years or more. Twenty-two percent (22%) of the participants reported to have 20 plus years of work experience in the organization, and 36% of the participants had 10–20 years of work experience. Most participants reported chronic illness such as sleep apnea/sleep disorder (58%), hypertension (46%), and cholesterol (42%) (Figure 2). Overreporting or underreporting of illnesses is possible in a self-reported survey [21]. However, this online study was designed so that participant’s personal identification was kept anonymous, which may have influenced participants to freely report the truth about their job stress information and health without fear of any consequences.

### 3.2. Job Stress Indicators

Study participants were recruited from October 2019 to July 2020. Job stress indicators like the effort–reward ratio (ERR) and overcommitment (OC) were determined from the survey questionnaire using the effort–reward imbalance model. ERR and OC values of participants during the study months are shown in Figure 3. In January 2020, the World Health Organization declared the coronavirus outbreak and the world became aware of COVID-19. High ERR scores were observed among participants in March, June, and July months of 2020, and high OC scores were observed among participants during those months, with March 2020 showing the highest average of OC scores compared to other participation months (Figure 3).

ERR values were divided into three groups: low (score 0.25–0.74), intermediate (score 0.75–1.00), and high (score 1.01–4.00) based on Siegrist et al. [18,19,20]. In the groups, 6% of the participants fell into the low ERR, 11% in the intermediate, and 83% in the high group. A high effort–reward ratio indicates that stress is observed in most EMS participants (83%). Overcommitment was also divided into three groups: low (score 6–11), intermediate (score 12–13), and high (score 14–24) with 3% of participants in low, 8% in intermediate, and 89% in high. Most participants (89%) in the group are highly overcommitted to their work (Figure 4).

Logistic regression analysis was performed to examine the independent effects of the variables age, BMI, sex, marital status, weekend work, continuous work hours, and job stress indicators—ERR and OC (all variables were categorical). The results were shown as odds ratio and 95% confidence interval. ERR > 1 (Range 1.01–4 score is High ERR) was found in 30/36 (83%) participants, indicating more effort and less reward, thus indicating stress. OC > 13 (Range 14–24 score is High OC) was found in 32/36 (89%) participants, indicating stress and future health risks. High ERR was associated with demographic and socioeconomic factors of participants (higher odds ratio) such as age of 40 years or more and continuous work of 24 h or more. High overcommitment was observed by high odds ratio in females, in married participants, and in continuous work of 24 h or more (Table 2 and Table 3).

### 3.3. Sleep Apnea, Shiftwork, and Job Stress Indicators

Significant correlation (*p* < 0.05) was found between job stress indicators ERR and OC and sleep apnea in participants (Table 4). Significant correlation (*p* < 0.05) was found between job stress indicators ERR and OC and sleep apnea in participants who work in rotating shifts. There were 10 day shift employees and 26 rotating shift employees who participated in our study. We found that psychosocial work environments may be associated with occupational stress among EMS, as revealed by high ERR, OC, and their significant correlation with sleep apnea in rotating shift employees (Table 4). ERR scores in rotating shift (mean 1.325, median 1.333, SD 0.325, range 0.686–1.974) were greater (*p* 0.221) than day shift (mean 1.188, median 1.200, SD 0.195, range 0.875–1.473) employees, although the difference between the two groups was not significant. OC scores in rotating shift employees (mean 16.307, median 16, SD 2.204, range 13–21) were greater (*p* 0.224) than day shift (mean 15.2, median 15, SD 2.573, range 10–20) employees, although the difference was not significant (Figure 5).

## 4. Discussion

Coping with stress and burnout is a common problem among emergency medical service professionals [2]. Emergency medical technicians (EMTs) and paramedics respond to emergency calls, performing medical services and transporting patients to medical facilities [22]. Most EMTs and paramedics work full time for 40 h or more per week. Some EMTs and paramedics work 12- or 24-hour shifts. Their work can be physically demanding and strenuous, and more often than not involving life-or-death circumstances [22]. The COVID-19 pandemic substantially increased the workload among these professionals and increased physical and mental distress during the COVID-19 pandemic, which has led to fatigue and burnout.

Burnout is a global health concern among EMT, paramedics, physicians, and nurses as well as other healthcare professionals [23]. There are several articles published on fatigue and burnout among healthcare professionals during COVID-19 [24,25]. A systematic review published in 2021 reported that chronic stress can lead to burnout causing fatigue, pessimism about job, and reduced professional effectiveness [26]. Since the onset of the COVID-19 pandemic, healthcare professionals have faced challenges including exposure to a wide range of occupational stressors such as high demand in job, higher workload, unusual workload, prolonged wearing of personal protective equipment (PPE), lack of sleep, etc. [27]. There are several effective techniques reported in previous studies to prevent or reduce burnout among healthcare employees, which include improving work schedules and work––life balance; effective organizational structure and flexibility; self-management skills; and training on physical, mental, emotional care, practicing mindfulness, and other stress-reducing techniques [28,29,30,31]. This study aimed to determine psychosocial stress levels in EMS professionals before and during COVID-19 and investigated stressors associated with work within the EMT population with high job demand.

Several factors lead to work stress among EMS personnel during the pandemic. A cross-sectional study from Spain published in October 2022 reported stress and burnout among EMTs during the pandemic [5]. Surveys of participants indicated 99% of participants had direct contact with COVID-19 patients, 54% did not have adequate personal protective equipment, 45% of the participants or their family members were diagnosed with COVID-19, and 63% of the study participants were afraid of getting infected with the virus [5]. The study findings suggested factors that may lead to stress and burnout were working during the pandemic without proper PPE, female and young age, early career level, and risk of getting infected [5]. Another study addressed the psychological burden on paramedics during COVID-19. Factors related to the stress and burden included facing ethical and moral dilemma to treat COVID patients without proper PPE, fear of contracting disease, loss of majority of the workforce amplified the burden on the remaining workers, and increased work hours and work demand [32]. The problems were faced even more in developing and underdeveloped countries who lacked resources to manage the pandemic, had shortage of trained healthcare workers, and had a greater deficiency of PPE [32]. A study conducted on Israeli paramedics leaving the profession was published in November 2019. The study reported contributing factors that lead to increase in quitting the profession were lack of career options, strenuous work, physical demands, inadequate salaries, long work hours, and shift work [33].

A study published in 2021 showed dynamic psychosocial risk and protective factors were associated with mental health in EMS personnel [34]. The psychosocial factors in the study consisted of occupational stressors, such as sleep disturbance, social conflict, social support, and so on [34]. Another study published from Spain in 2022 reported that, during COVID-19, more than half of the participating EMTs (53%) perceived a moderate stress level, 37% perceived moderate levels of emotional exhaustion (EE), and 40% had moderate levels of depersonalization (DP) [5]. The study reported COVID-19 pandemic had a tremendous impact on the mental health of ambulance EMTs [5]. Our study is aligned with findings from previous studies during COVID-19 indicating psychosocial stress among EMTs. Our study showed the psychosocial stress indicators effort–reward ratio is greater than 1 in most participants (83%), indicating an imbalance between effort and reward at work. EMS professionals exert more effort at work and get less reward from work, which can lead to psychosocial stress. Most participants (89%) were overly committed to their jobs. Females were more highly overcommitted to the job than males (high odds ratio), although the difference in OC scores between males and females were not significant. This could be attributed to the low sample size of our study. EMTs in our study participated in October of 2019 and December of 2019 before the world knew about COVID-19. Results showed participants had high ERR and high OC scores before COVID-19, indicating stress. Stress indicator ERR scores were high among participants in March, June, and July months of 2020 when the world was aware of COVID-19. Stress indicator OC scores were high among participants during those months, with March 2020 showing the highest average of OC scores compared to other participation months as shown in Figure 3. During those times, EMTs were busy worldwide assisting patients, comforting patients in difficult situations, and transporting them to hospitals, risking their own life and health to COVID-19 exposures. The unknown risks associated with COVID-19 during the onset of the disease lead to mental stress among EMTs, and the sudden rise of COVID-19 cases posed unique challenges that frontline workers had to deal with [35,36]. The high ERR and OC scores clearly indicated the crisis that EMS professionals faced as frontline workers during the onset of COVID-19 (February–March of 2020).

There is a difference in emergency medical services provided to patients globally. A difference exists among Asia-Pacific countries and European countries. EMS systems were single-tiered in Asia-Pacific countries, and most were public and fire-based, such as in Thailand, Malaysia, Singapore, Taiwan, Japan, and Korea. Ambulance personnel were primarily emergency medical technicians and paramedics, except for Thailand and Turkey, whose personnel include nurses and physicians [37]. Personnel were trained to use automated external defibrillators and have basic cardiac life support certification. The service capability of each EMS system varied greatly in terms of dispatch, airway management, and medications [37]. In the United States, EMS professionals provide out-of-hospital acute medical care and transport to definitive care for patients [38]. The EMS is regulated by the US National Highway Traffic Safety Administration. An EMS can be publicly or privately (for profit) operated. There is wide variation among states, and even among counties within states, of what type of care providers at different levels are allowed to provide. Depending on level of education and training, EMTs can administer CPR, glucose, and oxygen, and paramedics can perform more complex procedures such as inserting IV lines, administering drugs, and applying pacemakers [38].

Our study is in alignment with many previous research findings conducted in the U.S. and in different parts of the world. A study published in the year 2021 from Poland on EMS professionals during COVID-19 reported that stress among emergency medical service personnel increased during the pandemic due to new factors that previously did not exist, such as fear of contracting COVID-19, decrease in level of safety, and limitation of treatment for patients not suffering from COVID-19 [39]. Some sociodemographic factors such as being female and serving in the nursing profession are other reasons for stress as reported by the study [39]. Our study findings are similar in reporting higher stress among EMS personnel during the onset of COVID-19.

Another study published in 2021 performed a retrospective review of prehospital EMS responses from 22 urban, suburban, and rural EMS agencies in Western Pennsylvania of United States [15]. The study reported high prevalence of depression, anxiety, and stress among EMS personnel and paramedics during COVID-19. The study also reported that the married responders were likely at higher risk for depression and anxiety compared to unmarried responders. The study concluded that early assessment and management of mild depression, anxiety, and stress are crucial among first responders [15]. Our study is similar in reporting stress among EMS during COVID-19; however, it differs from this study finding, as we did not find higher risk of psychosocial stress among married responders compared to unmarried EMS participants.

A study from Taiwan in 2022 investigated psychological stress and sleep quality of emergency medical technicians in Taiwan Fire Department during the COVID-19 pandemic [40]. The study reported factors during the COVID-19 pandemic that contributed to increased depression, anxiety, and stress levels among EMS were poor sleep quality and a lack of understanding and support from the firefighting agency staff, family, or peers. [40]. Our study findings support the results that having sleep apnea, or a sleep disorder is found to be associated with stress in the EMS personnel.

Shift work, weekend work, and long work hours are contributors to health risk, as shown by previous studies [41,42]. Working night shift and long hours were reported to be the contributing factors for burnout syndrome in emergency professionals [13]. Our study in Table 2 and Table 3 showed high ERR and OC scores were associated with continuous work of a 24-h shift or more. High ERR was associated with participants with age 40 years or more, and high OC was associated with higher BMI (overweight and obese). However, these associations (odds ratio) were not significant (*p* > 0.05) likely due to the low sample size (n = 36) of our study.

In summary, psychosocial job stress was observed in most participants in our study. All efforts were made to minimize selection bias of the study by having inclusion criteria to include all participants working in the EMS agencies irrespective of difference in gender, race, BMI, and age. There were several reasons a cross-sectional study was chosen: the exploratory study gave the opportunity to investigate stress at a specific point of time and provided opportunity for future research, besides being affordable and less time consuming. This study had several limitations. First, it is a cross-sectional study design, so we were unable to demonstrate any chronobiological effects of work stressors. We could not determine any information on the causality of the job stress indicators and stressors/variables. Follow-up investigations over decades among these participants will inform any change in relationship between job stress indicators and other variables. In addition, the relatively small sample size (n = 36) may have limited the generalizability of the study. In the future, more participants from different EMS agencies could be recruited to expand this study. Multiple EMS agencies can be studied for identifying associations between psychosocial work stress and stressors in rural and urban areas.

EMS professionals are commonly exposed to long work hours as well as shift work [22]. Such demanding schedules often lead to sleep disorders because of the need to sleep at irregular times, resulting in irregular circadian rhythms, poor sleep quality, and shorter sleep duration [43]. Our study reported that many EMTs work a 24 h on/48 h off shift schedule in rural areas. Moreover, there is a social stigma for professionals who take naps at the workplace in some areas, which might contribute to sleep deprivation when working long hours. Epidemiological studies support increased risk of cardiovascular diseases among shift workers suffering from sleep disturbances [44]. We investigated the relationship between job stress indicator ERR and sleep apnea among EMTs working rotating shifts. We reported high ERR and high OC scores among participants, indicating stress was associated with sleep apnea in our study. We divided the EMTs based on their shiftwork into dayshift and rotating shift employees. We observed that high ERR and high OC scores were found to be associated with sleep apnea in rotating shift employees. This is supported by few studies that indicated sleep apnea and sleep disturbances were associated with rotating shiftwork. A study published in 2020 from Australia on paramedics reported higher level symptoms of depression, anxiety, fatigue, insomnia, and poorer sleep quality than the general population [45]. Another study published in 2007 reported that shiftwork can induce sleep apnea among patients and acute sleep deprivation may worsen the obstructive sleep apnea index [46]. Addressing sleep disorders and providing shift preference to EMS professionals may help to reduce depression and anxiety and improve mental and physical health [47].

There are several limitations of the study including small sample size, cross-sectional study design, and collecting surveys at a single point of time. The small sample size may limit the statistical power to detect statistically significant results in some variables, limiting the generalizability of the results. Binary logistic regression analysis was used for determining the effects of the independent variables such as, age, BMI, sex, marital status, weekend work, continuous work hours, and job stress indicators—ERR and OC. However, some variables with higher odds ratio did not have a significant *p* value due to the small sample size. Cross-sectional design was used to measure the prevalence of stress and health risk at a single point of time. With the cross-sectional study design, the absence of follow-up of participants might lead to inability to access incidence and make a causal inference. The preliminary evidence lays the groundwork for a large-scale study with involving more EMS agencies across geographical locations to further examine the impact of stress and health in the population and focus on interventions to eliminate stress. Having acknowledged these limitations, the findings of this feasibility study shed light on several insights that support the understanding that perceived psychosocial stress exists among EMS professionals. Findings suggested association between stress indicators, sleep apnea, rotating shifts, and long work hours.

Despite these limitations, the findings of the study reflect the association between the psychosocial work environment and work stressors in the EMS job. In the future, the findings of the study may be verified with large-scale and multi-center longitudinal studies involving many EMS agencies in rural and urban areas. Several studies have focused on the role of effective interventions to reduce stress among healthcare professionals [26,48,49]. The practice of yoga and mindfulness have been shown to improve the physical, emotional, and mental health of healthcare workers via reduction of stress and burnout [49,50]. Investigating effective interventions among EMS agencies to reduce stress and burnout can be beneficial to both EMTs and their patients. Studies have shown that EMS providers experienced many challenges during pandemic as frontline workers. However, there is still a scarcity of research focusing on EMS providers psychological challenges during COVID-19 in many parts of the world [51]. Our study showed prevalence of psychosocial stress and health risk in EMS professionals working around the clock in EMS agencies before and during pandemic. More research may be conducted on EMS personnel mental health perspectives and needs before and during pandemic [51]. This type of research will help prepare the EMS work force for any future pandemic based on the lessons learned. Research on effective interventions to meet EMS personnel mental health needs is significant to manage crisis and improve the quality of the treatment provided by the EMS professionals. In future, more research needs to focus on mental health challenges faced by EMS professionals around the world and effective interventions to improve the mental health of EMTs and paramedics. The findings of these research studies may help manage crisis and improve the quality of patient care provided by the EMS professionals. Interventions that focus on personal improvement and organizational change may benefit EMS personnel to cope better with stress and burnout in crisis.

Although several limitations exist in our study, the study is significant in understanding the psychosocial work environment before and during the COVID-19 pandemic in EMS agencies as well as the possible associations of psychosocial job stress and health risk factors, such as working long hours and rotating shifts, in EMS professionals.

## 5. Conclusions

Emergency medical professionals played an important role during the onset of the COVID-19 pandemic, working as first responders on the frontline [2]. EMS professionals already faced many challenges before the COVID-19 pandemic, including work overload, night shifts, and irregular working hours, which further escalated during the pandemic [2,5,22]. Our study reported psychosocial work stress among emergency medical professionals before and during COVID-19 as revealed by high effort–reward ratio and high overcommitment scores. However, a longitudinal study involving more EMS agencies should be conducted to strengthen the association between psychosocial work stress and occupational stressors in EMTs. This study provides meaningful knowledge in psychosocial stress and risk factors during the COVID-19 pandemic, strengthening the need for EMTs’ psychological preparedness and improving the quality of preventive measures.

## Figures and Tables

**Figure 1 healthcare-11-00976-f001:**
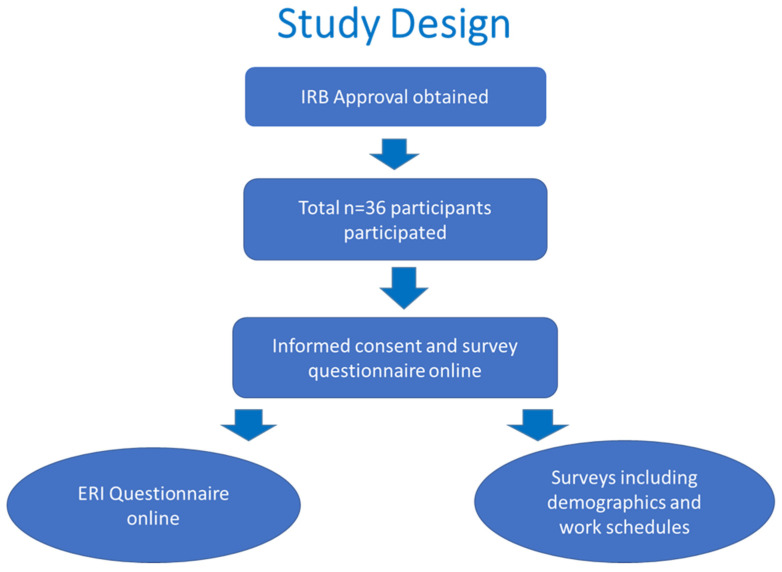
A flow diagram showing the study design.

**Figure 2 healthcare-11-00976-f002:**
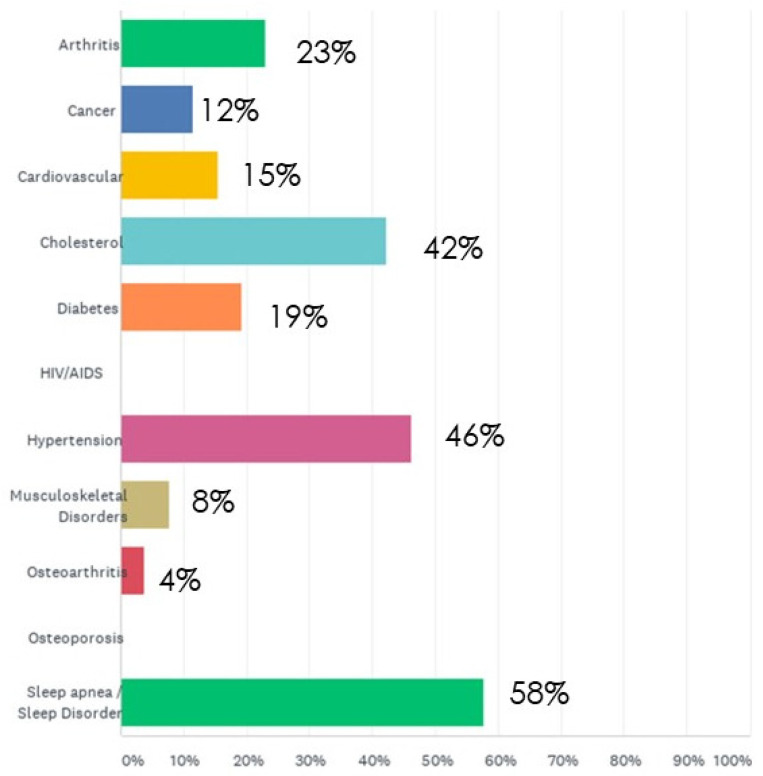
A graph reporting participant’s chronic disease from their survey. Participants filled out survey questionnaires providing information on their chronic diseases. Out of the 36 participants (n = 36), the majority of the participants reported having sleep apnea, hypertension, and cholesterol. From the survey findings, it was found that 58% participants had sleep apnea/sleep disorder, 46% had hypertension, and 42% had cholesterol.

**Figure 3 healthcare-11-00976-f003:**
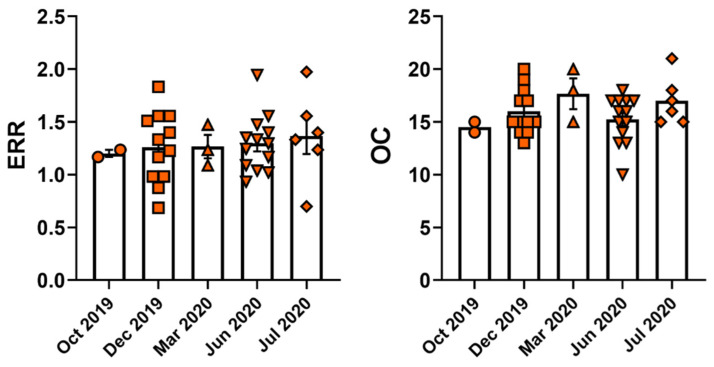
ERR and OC values calculated from participant’s survey during the months of 2019–2020. The study was conducted from October 2019 to July 2020. Job stress indicators: ERR and OC were calculated from participant’s survey questionnaires using the effort–reward imbalance model. A total of 36 participants participated (n = 36). In October 2019—2 participants, December 2019—12 participants, March 2020—3 participants, June 2020—13 participants, and July 2020—6 participants participated in our study. High ERR (1.166–1.974) and high OC (14–21) scores were observed in most participants before and during COVID-19 in the years 2019 and 2020.

**Figure 4 healthcare-11-00976-f004:**
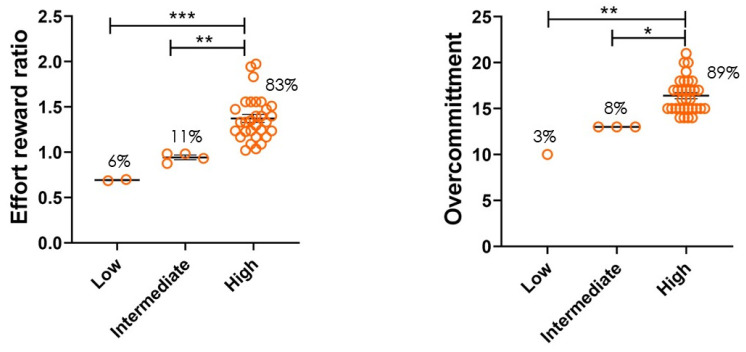
ERR and OC scores among emergency medical service professionals. Participants filled out survey questionnaires based on the effort–reward imbalance model. Job stress indicators: ERR and OC scores were calculated. ERR and OC scores were divided into low, intermediate, and high ranges calculated from the participant survey. High ERR scores were observed in 83% of participants, and high OC scores were observed in 89% of participants. *** *p* < 0.001, ** *p* < 0.01, * *p* < 0.05. Difference in ERR Low to High group is highly significant (*** *p* < 0.001) and ERR Intermediate to High group is also significantly different (** *p* < 0.01) as shown in graph, Significant difference is also found in OC Low to High and OC Intermediate to High group.

**Figure 5 healthcare-11-00976-f005:**
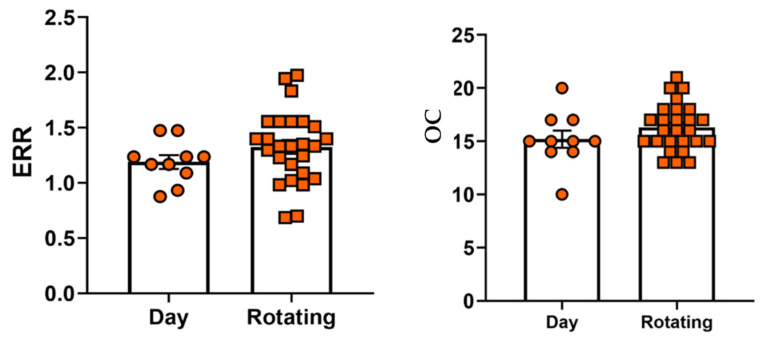
ERR and OC scores were shown in charts with individual data points in day and rotating shift emergency medical service professionals.

**Table 1 healthcare-11-00976-t001:** Descriptive statistics of the study participants.

Characteristics	Total (n = 36) n% or Mean (SD) *	Median (Range)
Age (years) *	39.028 (10.722)	39 (20–65)
Sex		
Female	10 (28)	
Male	26 (72)	
BMI *	31.633 (7.863)	30.2 (18–56.1)
Marital Status		
Married	23 (64)	
Single	13 (36)	
Shiftwork		
Dayshift	10 (28)	
Rotating shift	26 (72)	
Weekend Work		
Yes	31 (86)	
No	5 (14)	
Continuous Work hours/week		
≤12	15 (42)	
≥24	21 (58)	
ERR *	1.287 (0.298)	1.270 (0.686–1.97)
OC *	15.944 (2.254)	15.5 (10–21)

* Standard deviation (SD) values for age, body mass index (BMI), ERR, and OC.

**Table 2 healthcare-11-00976-t002:** Demographic and socioeconomic indicators as risk factors for high ERR.

Characteristics	OR (95% CI)	*p* Value
Age (years)		
≤40	1.00 (Reference)	
>40	3.002 (0.265–33.970)	0.374
Sex		
Female	0.355 (0.012–10.370)	0.547
Male	1.00 (Reference)	
Marital Status		
Married	0.000 (0.000)	0.998
Single	1.00 (Reference)	
Weekend Work		
Yes	0.000 (0.000)	0.999
No	1.00 (Reference)	
Continuous Work hours/week		
≤12	1.00 (Reference)	
≥24	12.565 (0.223–709.28)	0.219
BMI		
Obese (≥30)	0.000 (0.000)	0.999
Overweight (25.0–29.9)	0.000 (0.000)	0.999
Normal (<25)	1.00 (Reference)	

**Table 3 healthcare-11-00976-t003:** Demographic and socioeconomic indicators as risk factors for high OC.

Characteristics	OR (95% CI)	*p* Value
Age (years)		
≤40	1.00 (Reference)	
>40	0.305 (0.015–6.232)	0.440
Sex		
Female	1.317 (0.068–25.629)	0.856
Male	1.00 (Reference)	
Marital Status		
Married	1.835 (0.114–29.589)	0.669
Single	1.00 (Reference)	
Weekend Work		
Yes	0.000 (0.000)	0.999
No	1.00 (Reference)	
Continuous Work hours/week		
≤12	1.00 (Reference)	
≥24	2.296 (0.161–32.804)	0.540
BMI		
Obese (≥30)	8.221 (0.210–321.406)	0.260
Overweight (25.0–29.9)	4.723 (0.129–172.821)	0.398
Normal (<25)	1.00 (Reference)	

**Table 4 healthcare-11-00976-t004:** Showing correlation between rotating shift and sleep apnea in participants.

Variables	Pearson r	95% CI	R Squared	*p* (Two Tailed)
ERR vs. Sleep Apnea	0.3949	0.07623–0.6403	0.1559	0.0172 *
OC vs. Sleep Apnea	0.5534	0.2748–0.7463	0.3063	0.0005 *
ERR vs. Sleep Apnea in Rotating Shift Employees	0.5731	0.2387–0.7860	0.3284	0.0022 *
OC vs. Sleep Apnea in Rotating Shift Employees	0.5623	0.2236–0.7798	0.3161	0.0028 *

* Significant if *p* value < 0.05.

## Data Availability

The datasets generated during and/or analyzed during the current study are available from the corresponding author upon reasonable request.

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
