# Peer review of "Psychosocial Work Stress and Occupational Stressors in Emergency Medical Services"

_healthcare, 2023, doi:10.3390/healthcare11070976_

Round 1

Reviewer 1 Report

A study was conducted to identify psychosocial work stress among emergency medical services providers. The Effort Reward Imbalance model was used to identify two job stress indicators: Effort Reward Ratio (ER Ratio) and Over- commitment (OC). Overall, the study is well planned out and the findings are of high interest, especially for the continuous improvement of welfare among medical service providers. The main concern of this paper would be the low count of sample size. Although this is addressed as the limitations of the study, it is important to be aware that the implication it has on the overall trustworthiness of the research. Other minor mistakes and comments are provided in the attached file. 

Reviewer 2 Report

I enjoyed reading your work. Here are some of my concerns:

1. I understand that due to the situation, the study was carried out with the data obtained from the people willing to answer the instrument, but would it be possible to add a sample sufficiency test to the study?

2. How do you limit your results in conducting a study based on such a small sample?

3. The abstract is a very important slice of the article; it decides if the reader will continue reading your paper or not. I recommend you improve it a little more.

4. I recommend improving the Introduction session, since the importance of developing the research is not enough to seem.

5. The Discussion should be strengthened, making more comparisons between similar studies in other countries or different types of medical services.

6. I would love to see more suggestions for future work (if possible).

7. Finally, I recommend increasing the references in your article, and that they focus on the causes of work stress in emergency medical services personnel.

Round 2

Reviewer 2 Report

I believe that the authors adequately addressed my concerns and considerably improved the quality of the manuscript.